# Effect of Li Content on the Microstructure and Mechanical Properties of as-Homogenized Mg-Li-Al-Zn-Zr Alloys

**Yuehua Sun** [1,2]**, Fan Zhang** [1]**, Jian Ren** [1,2,*] **and Guangsheng Song** [1,2]

[1]  Key Laboratory of Green Fabrication and Surface Technology of Advanced Metal Materials (Anhui University of Technology), Ministry of Education, Maanshan 243002, China; sunyuehua1008@126.com (Y.S.)
[2]  School of Materials Science and Engineering, Anhui University of Technology, Maanshan 243002, China
*  Correspondence: renjianahut@126.com

**Abstract:** The microstructure and mechanical properties of as-homogenized Mg-*x*Li-3Al-2Zn-0.2Zr alloys (*x* = 5, 7, 8, 9, 11 wt.%) were studied. As the Li content increased from 5 wt.% to 11 wt.%, the alloy matrix changed from the α-Mg single-phase to α-Mg+β-Li dual-phase and then to the β-Li single-phase. Homogenized With the increase in Li content, the alloy strength decreased while the elongation increased, and the corresponding fracture mechanism changed from cleavage fracture to microvoid coalescence fracture. This is mainly attributed to the matrix changing from α-Mg with hcp structure to β-Li with bcc structure. Additionally, the increase in the AlLi softening phase led to the reduction of Al and Zn dissolved in the alloy matrix with increasing Li content, which is one of the reasons for the decrease in alloy strength.

**Keywords:** Mg-Li alloy; microstructure; mechanical property; fracture mechanism





## 1. Introduction

Alloying Mg with Li can produce the lightest Mg-Li alloy; its density (1.25–1.65 g/cm$^3$) is close to that of plastic, with only 3/5–3/4 of that of conventional Mg alloy and 1/2–2/3 of that of Al alloy. The Mg-Li alloy exhibits a unique phase transition with the change in Li content, i.e., the alloy is an α-Mg single-phase alloy with an hcp structure when the Li content is lower than 5.7 wt.%, the alloy is a β-Li single-phase with a bcc structure when the Li content is higher than 10.3 wt.%, and the alloy is an α-Mg+β-Li dual-phase alloy with an hcp+bcc structure when the Li content is between 5.7 wt.% and 10.3 wt.% [1]. The addition of Li can reduce the c/a axial ratio of the Mg lattice and even change the crystal structure from an hcp structure to a bcc structure, which makes the Mg-Li alloy exhibit incomparable plastic deformation ability compared to a conventional Mg alloy [1,2]. In addition, the Mg-Li alloy also shows the advantages of high specific strength, excellent damping, good electromagnetic shielding performance, and weak anisotropy, being lightweight and having good prospects in the fields of energy saving, and emission reduction, especially in aerospace and electron industries [3,4]. Taking aerospace satellites as an example, the application of the Mg-Li alloy can achieve a weight reduction effect of about 20–50%, significantly improving the payload of the satellite, and thus generating huge economic benefits.

However, since the binary Mg-Li alloy exhibits low strength, it is necessary to add alloying elements to strengthen the binary Mg-Li alloy. Al and Zn are the common alloying elements for a Mg-Li alloy; they can improve the strength of the Mg-Li alloy by solid solution strengthening and second phase strengthening, owing to their high solid solubility in the Mg matrix and ease of combining with other elements to form compounds [5]. Previous research has shown that Al and Zn have similar effects on Mg-Li alloy; that is, the alloy strength increases while the elongation decreases with the increase in Al content or Zn content [5,6]. In the Mg-Li-Al alloy, the Al content is generally less than 5 wt.%–6 wt.%, above which the alloy strength does not increase significantly and the elongation loss is

obvious. When the Zn content exceeds 2.5 wt.%, it has a negative effect on the corrosion resistance of the Mg-Li alloy, so the Zn content is usually controlled within 2.5 wt.% [7]. Zr is often used as a grain refiner to improve the microstructure and mechanical properties of an alloy, and its addition amount is generally 0.1 wt.%–0.3 wt.%. In recent years, due to the excellent mechanical properties of Mg-Li-Al-Zn (LAZ) alloys, much research has have focused on improving the mechanical properties of LAZ532 and LAZ832 alloys through alloying with rare earth elements [8–11]. Cui et al. [8] revealed that the addition of Y element into the LAZ532 alloy could form an $Al_2Y$ strengthening phase, reduce the AlLi softening phase, and refine grains, and the tensile strength was increased by 32.66% when the Y content was 0.8 wt.%. Zhu et al. [9] indicated that Y and Nd had a synergistic strengthening effect on the LAZ532 alloy, and the grain refinement effect was the best and the mechanical properties reached their maximum values (UTS = 231 MPa, δ = 16%) when adding 1.2 wt.% Y and 0.8 wt.% Nd simultaneously. Zhao et al. [10] showed that the LAZ832-0.5Y alloy had the best tensile strength of 218.5 MPa and elongation of 16.9%, which was mainly attributed to the grain refinement and second phase strengthening effect caused by the addition of Y. However, there are few systematic studies on the microstructure and mechanical properties of Mg-Li-Al-Zn alloys with different Li contents.

In this work, the microstructure and mechanical properties of as-homogenized Mg-$x$Li-3Al-2Zn-0.2Zr alloys ($x$ = 5, 7, 8, 9, 11 wt.%) were investigated, and the fundamental reason for the variation of mechanical properties was explained.

## 2. Experimental Procedures

The alloys used in this work were prepared by melting pure Mg (>99.9 wt.%), pure Li (>99.9 wt.%), pure Al (>99.9 wt.%), pure Zn (>99.9 wt.%), and the Mg-Zr (30 wt.%) master alloy in a vacuum induction melting furnace under pure argon atmosphere. The melts were held at 730 °C for 10 min, and then poured into a stainless steel mold with a diameter of 80 mm and cooled to room temperature. After that, the cast ingots were homogenized at 280 °C for 24 h to obtain the experimental alloys with uniform composition and microstructure. Choosing the homogenization temperature of 280 °C not only ensured that the alloys could be completely homogenized, but also ensured that the alloys were kept warm for a long time without any combustion hazards. The densities of as-homogenized alloys were determined with an electronic density balance (AEL-200, Mettler Toledo, Greifensee, Switzerland). The phase composition was analyzed using an X-ray diffractometer (XRD, D/Max 2500, Rigaku, Tokyo, Japan) using monochromatic Cu Kα radiation, and the 2θ was in the range of 20° to 80° with a step size of 0.02° and a scan rate of 4°/min. The microstructure was observed with a scanning electron microscope (SEM, Quanta-200, FEI, Eindhoven, The Netherlands), an electron probe microanalyser (EPMA, JXA-8230, JEOL, Tokyo, Japan), and a transmission electron microscope (TEM, Tecnai G20ST, FEI, Eindhoven, The Netherlands). The samples for TEM observation were firstly ground to 80 μm, then thinned to perforation with a twin-jet electropolisher, and finally thinned for 20 min using an ion-beam thinner at a small angle. The electrolyte for twin-jet electropolishing was a mixed solution containing 8 vol.% perchloric acid and 92 vol.% alcohol, and the specific experimental parameters were a temperature of about −30 °C, working voltage of 12–20 V, and working current of 12–16 mA. The tensile tests were carried out with a tensile tester (MTS-858, MTS, Costa Mesa, CA, USA) at a tensile rate of 2 mm/min, and the dimensions of samples for tensile tests were 30 mm in gauge length and 6 mm in diameter. The hardness was measured with Vickers hardness tester (HV-50AP, Mitutoyo, Kanagawa, Japan) with a loading force of 98 N and a holding time of 10 s.

## 3. Results and Discussion

### 3.1. Microstructure

Figure 1 shows the XRD patterns of as-homogenized Mg-$x$Li-3Al-2Zn-0.2Zr alloys. Among these alloys, the LAZ532-0.2Zr alloy is a typical α-Mg single-phase alloy with an hcp structure, while the LAZ1132-0.2Zr alloy is a typical β-Li single-phase alloy with a

bcc structure. When the Li contents were 7 wt.%, 8 wt.%, and 9 wt.%, the alloys were α-Mg+β-Li dual-phase alloys with an hcp+bcc mixed structure. Obviously, as the Li content increased from 5 wt.% to 11 wt.%, the matrix phase of the alloy changed from α-Mg single-phase to α-Mg+β-Li dual-phase and then to β-Li single-phase, and the crystal structure transformed from hcp structure to hcp+bcc structure and then to bcc structure, which are results consistent with the phase diagram of the Mg-Li binary alloy [12]. Except for the matrix phase, all alloys contained the AlLi phase (fcc structure), and the corresponding phase peaks become more and more obvious with increasing Li content. No phases containing Zn and Zr elements were detected in the alloys, which indicates that Zn and Zr exist in the alloy matrix in the form of solid solutions and do not form intermetallic compounds.

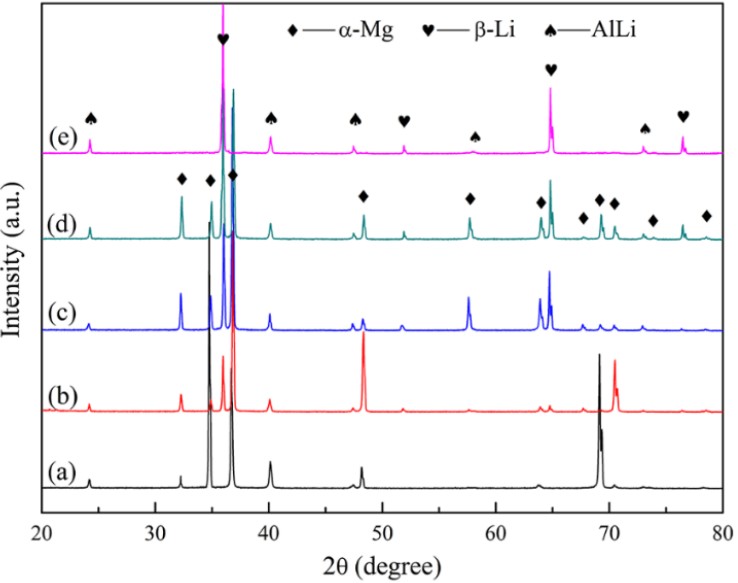

**Figure 1.** XRD patterns of as-homogenized Mg-*x*Li-3Al-2Zn-0.2Zr alloys: (**a**) *x* = 5; (**b**) *x* = 7; (**c**) *x* = 8; (**d**) *x* = 9; (**e**) *x* = 11.

Figure 2 displays the SEM micrographs of as-homogenized Mg-*x*Li-3Al-2Zn-0.2Zr alloys. The as-homogenized LAZ532-0.2Zr alloy was composed of a light gray α-Mg matrix phase, with a kind of white fibrous phase in the α-Mg matrix. According to the XRD results (Figure 1) and works in the literature [13,14], this fibrous phase can be identified as the AlLi phase, and it exists in the α-Mg matrix in the form of eutectic. As-homogenized LAZ732-0.2Zr, LAZ832-0.2Zr, and LAZ932-0.2Zr alloys consisted of a light gray α-Mg matrix and a dark gray β-Li matrix. With the increase in Li content, the α-Mg phase changes from a smooth block to long strip, and its content decreases gradually. It can be seen from the local enlarged view in the lower left corner of Figure 2c that a white fibrous phase similar to that in Figure 2a is distributed in the α-Mg matrix and a white granular phase is distributed in the β-Li matrix. According to the XRD results (Figure 1) and the literature [15,16], both the fibrous phase in the α-Mg matrix and the granular phase in the β-Li matrix were the AlLi phase. The as-homogenized LAZ1132-0.2Zr alloy consisted entirely of a dark gray β-Li matrix phase, and a large number of AlLi particles were uniformly distributed in the β-Li matrix.

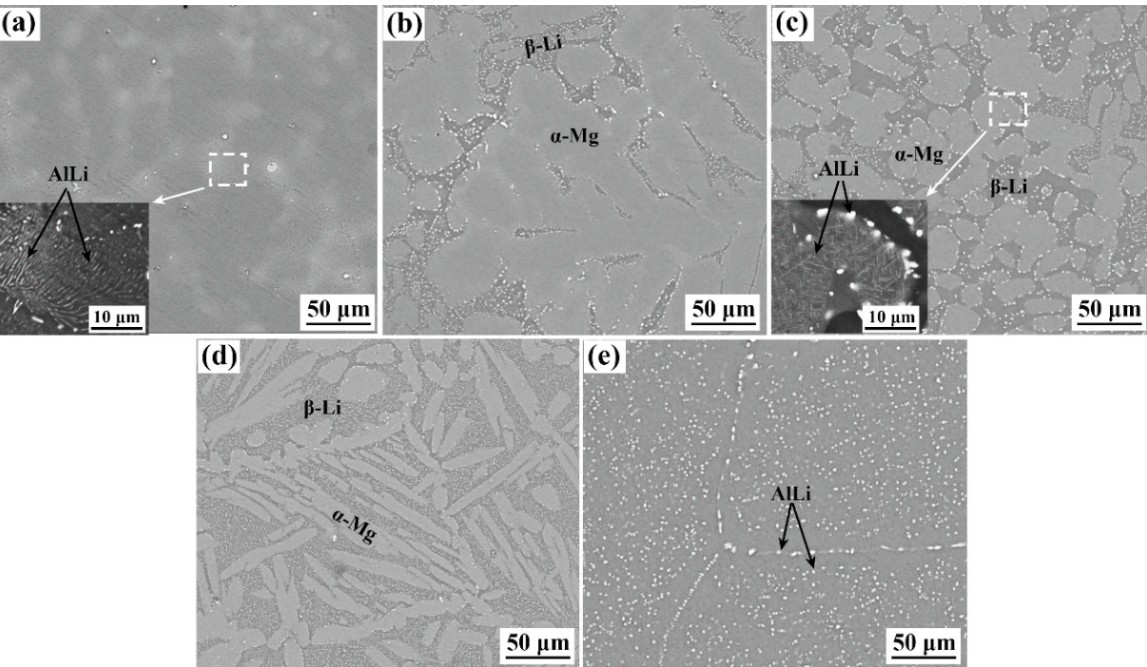

**Figure 2.** SEM micrographs of as-homogenized Mg-*x*Li-3Al-2Zn-0.2Zr alloys: (**a**) *x* = 5; (**b**) *x* = 7; (**c**) *x* = 8; (**d**) *x* = 9; (**e**) *x* = 11.

Figure 3 depicts the element distribution of as-homogenized LAZ532-0.2Zr, LAZ832-0.2Zr, and LAZ1132-0.2Zr alloys. Clearly, both the fibrous AlLi phase and the granular AlLi phase were rich in Al and Zn, and poor in Mg. It is worth noting that Li is too light to detect. The enrichment of Zn in the AlLi phase was mainly due to the high solid solubility of Zn in Al (83.1 wt.%) [17]. A part of the added Al element was combined with Li to form the AlLi phase, and the remainder was dissolved in the alloy matrix. Because the solid solubility of Al in Mg (12.7 wt.%) is much higher than that of Al in Li (extremely limited), the dissolved Al element was mainly distributed in the α-Mg matrix. A part of the added Zn element was enriched in the AlLi phase, and the remainder was dissolved in the alloy matrix. Because the solid solubility of Zn in Mg (6.2 wt.%) is lower than that of Zn in Li (12.5 wt.%), the solid solution concentration of Zn in the β-Li matrix was higher than that in the α-Mg matrix, but the difference was not great.

Figure 4 displays the TEM micrographs of the AlLi phase in the as-homogenized LAZ832-0.2Zr alloy. The AlLi phase distributed in the α-Mg matrix and β-Li matrix shows quite different morphologies. Fibrous AlLi phases with a width in the range of 0.02–0.11 μm are found in the α-Mg matrix, and round-like AlLi particles with a diameter in the range of 0.42–0.92 μm are observed in the β-Li matrix, which is consistent with the microstructure shown in Figures 2 and 3.

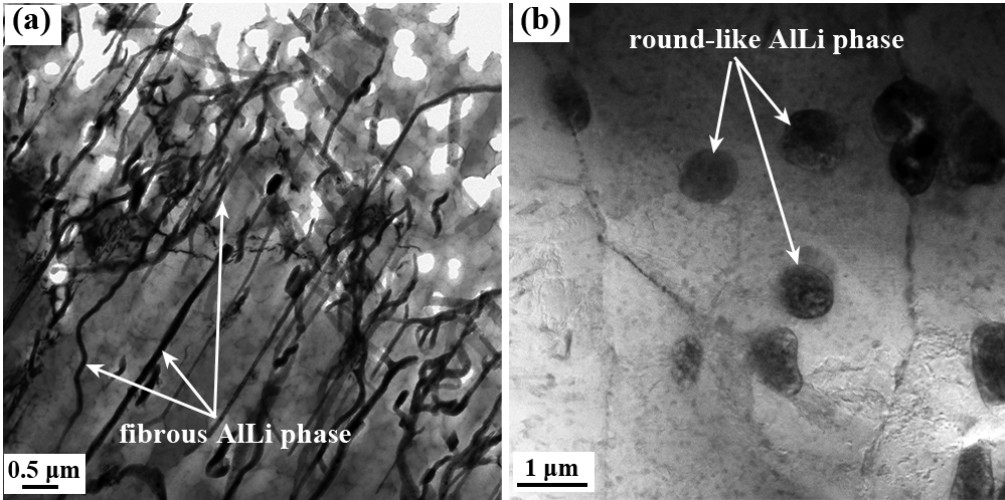

**Figure 3.** EPMA area analysis of (**a**) LAZ532-0.2Zr, (**b**) LAZ832-0.2Zr, and (**c**) LAZ1132-0.2Zr alloys.

**Figure 4.** TEM micrographs of (**a**) fibrous AlLi phase and (**b**) round-like AlLi phase in as-homogenized LAZ832-0.2Zr alloy.

### 3.2. Mechanical Properties

Figure 5 shows the stress–strain curves and hardness of as-homogenized Mg-$x$Li-3Al-2Zn-0.2Zr alloys, and the densities and mechanical properties of the alloys are listed in Table 1. The ultimate tensile strengths (UTS) of as-homogenized LAZ532-0.2Zr, LAZ732-0.2Zr, LAZ832-0.2Zr, LAZ932-0.2Zr, and LAZ1132-0.2Zr alloys were consecutively 197.2, 185.4, 181.4, 168.7, and 143.7 MPa; their elongations ($\delta$) were consecutively 10.1%, 11.6%, 21.5%, 24.7%, and 27.3%; and their hardness was consecutively 68.6, 64.1, 61.3, 58.9, and 56.2 HV. In addition, the specific strength decreased from 124.8 MPa/(g/cm$^3$) to 100.5 MPa/(g/cm$^3$) as the Li content increased from 5 wt.% to 11 wt.%, exhibiting excellent specific strength. Clearly, as the Li content increases, the strength and hardness of the alloy decreases while the elongation increases, which is consistent with the variation law reported in previous research [18,19]. This result is mainly attributed to the transformation of the crystal structure of the Mg-Li alloy from the hcp structure to the bcc structure. It has been reported that the hardness of the $\alpha$-Mg matrix is higher than that of the $\beta$-Li matrix [20]. With the increase in Li content, the content of the soft $\beta$-Li matrix in the alloy increased gradually, and thus the hardness of the alloy decreased gradually. In the Mg-Li alloy, the hard $\alpha$-Mg matrix with hcp structure showed higher strength and worse ductility, while the soft $\beta$-Li matrix with bcc structure showed lower strength and excellent ductility. For the dual-phase Mg-Li alloy, plastic deformation occurred preferentially in the soft $\beta$-Li matrix during the deformation process, and then the hard $\alpha$-Mg matrix began to undergo plastic deformation when the stress transmitted from the $\beta$-Li matrix to the $\alpha$-Mg matrix was greater than the elastic limit of the $\alpha$-Mg matrix. This is because the $\beta$-Li matrix with bcc structure is softer and has more independent slip systems than the $\alpha$-Mg matrix. Table 2 lists the independent slip systems and the critical resolved shear stress (CRSS) in Mg and Li at room temperature [21–24]. The common slip systems in Mg are $\langle \vec{a} \rangle$ basal slip, $\langle \vec{a} \rangle$ prismatic slip, $\langle \vec{a} \rangle$ pyramidal slip, and $\langle \vec{c} + \vec{a} \rangle$ pyramidal slip. The slip direction of these $\langle \vec{a} \rangle$ dislocation slips is $\langle 11\bar{2}0 \rangle$ direction perpendicular to the c axis, which cannot coordinate the strain in the c axis, while $\langle \vec{c} + \vec{a} \rangle$ dislocation slip along $\langle 11\bar{2}3 \rangle$ direction can coordinate the strain in the c axis. At room temperature, the CRSS required for the basal slip initiation of Mg is about 0.45–0.81 MPa, and the CRSS required for prismatic slip and pyramidal slip is about 100 times that required for basal slip. Therefore, the basal slip is the easiest to initiate during the deformation process of Mg at room temperature, but the prismatic slip and pyramidal slip are difficult to initiate. However, the basal slip of Mg can only provide two independent slip systems, which fails to meet the requirements of five independent slip systems in Von Mises criterion, and thus Mg has poor plasticity and is difficult to deform. Due to the limited slip systems of Mg, twinning often occurs to coordinate the c-axis strain during deformation at room temperature. Compared with compression twins ($\{10\bar{1}1\}$ and $\{10\bar{1}2\}$), tension twin ($\{10\bar{1}2\}$) is the most easily initiated twinning mode in Mg at room temperature because of its lower CRSS (2.0–2.8 MPa) and smaller shear displacement [25]. For Li with a bcc structure, the primary slip system is the $\{110\}\langle 111 \rangle$ with the secondary slip systems of $\{112\}\langle 111 \rangle$ and $\{123\}\langle 111 \rangle$. It has been reported that the $\{110\}\langle 111 \rangle$ slip system of Li has a lower CRSS (0.54–0.57 MPa), and the corresponding number of the independent slip system is 12, so Li exhibits good ductility during the deformation process [23]. Moreover, it has been considered that the excellent plasticity of the $\beta$-Li matrix is attributed to the nearly equal CRSS values of the primary slip system ($\{110\}\langle 111 \rangle$) and secondary slip systems ($\{112\}\langle 111 \rangle$ and $\{123\}\langle 111 \rangle$) [24]. Therefore, as the Li content increases, the content of the $\beta$-Li matrix with bcc structure increases gradually, and accordingly the tensile strength decreases while the elongation increases. In addition, as the Li content increases, the AlLi softening phase increases gradually, as shown in Figure 1, and the contents of Al and Zn dissolved in the alloy matrix decrease, which is one of the reasons for the gradual decrease in the alloy strength, although the effect is not great.

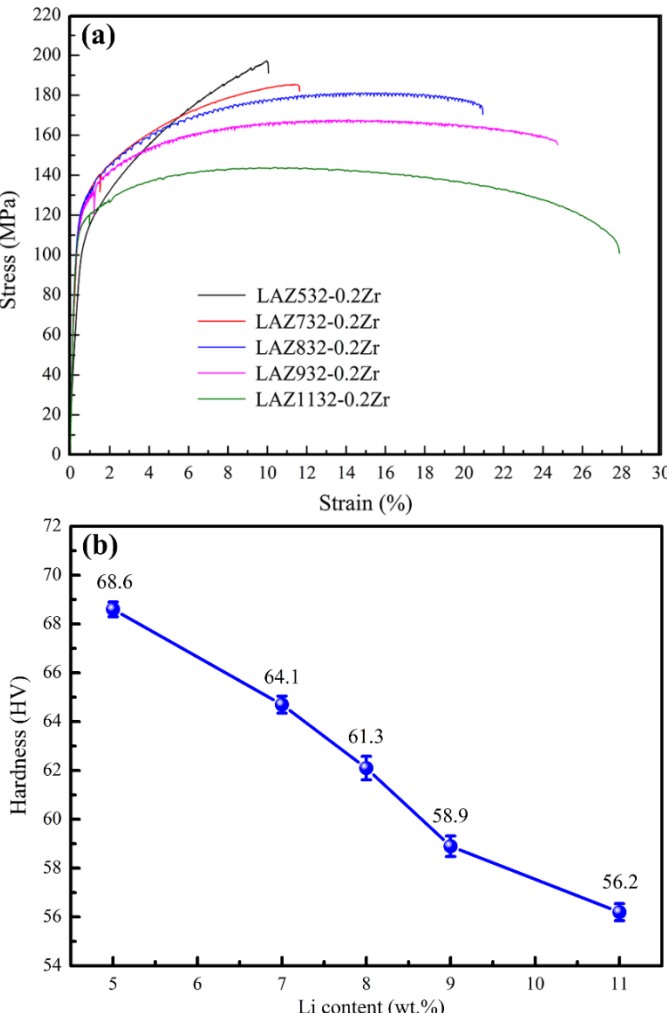

**Figure 5.** (**a**) Stress–strain curves and (**b**) hardness of as-homogenized Mg-*x*Li-3Al-2Zn-0.2Zr alloys.

Figure 6 depicts the fracture morphologies of as-homogenized Mg-*x*Li-3Al-2Zn-0.2Zr alloys. The fracture surface of the as-homogenized LAZ532-0.2Zr alloy was full of cleavage planes and cleavage steps, showing typical cleavage fracture characteristics. Cleavage fracture is caused by the stacking of dislocation at the grain boundary or twin boundary, which usually occurs along the cleavage plane, and the main cleavage planes of the α-Mg matrix with hcp structure were {0001} and {1$\bar{1}$00} with low index. In addition to cleavage planes and cleavage steps, many dimples could be observed on the fracture surfaces of as-homogenized dual-phase alloys (LAZ732-0.2Zr, LAZ832-0.2Zr, and LAZ932-0.2Zr), and the cleavage steps decreased while the dimples increased with the increase in Li content. Because a dimple is the basic feature of the microvoid coalescence fracture, the fracture modes of as-homogenized dual-phase alloys are mixture fractures of cleavage fracture and microvoid coalescence fracture. In the early stage of microvoid coalescence fracture, the microvoids are formed by the fracture of the second phase (AlLi), the separation of the second phase from the matrix, and the separation of the phase interface, owing to the elastic and plastic difference between the second phase and the matrix. The fracture surface of the as-homogenized LAZ1132-0.2Zr alloy was full of dimples with almost no cleavage steps, so its fracture mode was a microvoid coalescence fracture. The fracture mode of the as-homogenized alloys changed from cleavage fracture to microvoid coalescence fracture with increasing Li content, indicating that the elongation of the alloy gradually became better, which is consistent with the variation trend of elongation in Figure 5a.

**Table 1.** Density and mechanical properties of as-homogenized Mg-*x*Li-3Al-2Zn-0.2Zr alloys.

| Alloy | Density (g/cm$^3$) | UTS (MPa) | δ (%) | Hardness (HV) | Specific Strength (MPa/(g/cm$^3$)) |
|---|---|---|---|---|---|
| LAZ532-0.2Zr | 1.58 | 197.2 | 10.1 | 68.6 | 124.8 |
| LAZ732-0.2Zr | 1.53 | 185.4 | 11.6 | 64.1 | 121.2 |
| LAZ832-0.2Zr | 1.51 | 181.4 | 21.5 | 61.3 | 120.1 |
| LAZ932-0.2Zr | 1.48 | 168.7 | 24.7 | 58.9 | 114.0 |
| LAZ1132-0.2Zr | 1.43 | 143.7 | 27.3 | 56.2 | 100.5 |

**Table 2.** Independent slip systems and the CRSS in Mg and Li at room temperature [21–24].

| Metal | Type of Slip Plane | Type of Slip Direction | Slip Plane | Slip Direction | CRSS (MPa) | Number of Independent Slip Systems |
|---|---|---|---|---|---|---|
| Mg (hcp) | basal slip | $\langle \vec{a} \rangle$ | $\{0001\}$ | $\langle 11\bar{2}0 \rangle$ | 0.45–0.81 | 2 |
| | prismatic slip | $\langle \vec{a} \rangle$ | $\{1\bar{1}00\}$ | $\langle 11\bar{2}0 \rangle$ | 39.2–40 | 2 |
| | | $\langle \vec{a} \rangle$ | $\{11\bar{2}0\}$ | $\langle 11\bar{2}0 \rangle$ | | |
| | pyramidal slip | $\langle \vec{a} \rangle$ | $\{10\bar{1}1\}$ | $\langle 11\bar{2}0 \rangle$ | 45–81 | 4 |
| | | $\langle \vec{c} + \vec{a} \rangle$ | $\{11\bar{2}1\}$ | $\langle 11\bar{2}3 \rangle$ | | 5 |
| | | $\langle \vec{c} + \vec{a} \rangle$ | $\{11\bar{2}2\}$ | $\langle 11\bar{2}3 \rangle$ | | |
| Li (bcc) | prismatic slip | $\langle \vec{c} + \vec{a} \rangle$ | $\{110\}$ | $\langle 111 \rangle$ | 0.54–0.57 | 12 |
| | pyramidal slip | $\langle \vec{c} + \vec{a} \rangle$ | $\{112\}$ | $\langle 111 \rangle$ | — | 12 |
| | | $\langle \vec{c} + \vec{a} \rangle$ | $\{123\}$ | $\langle 111 \rangle$ | — | 24 |

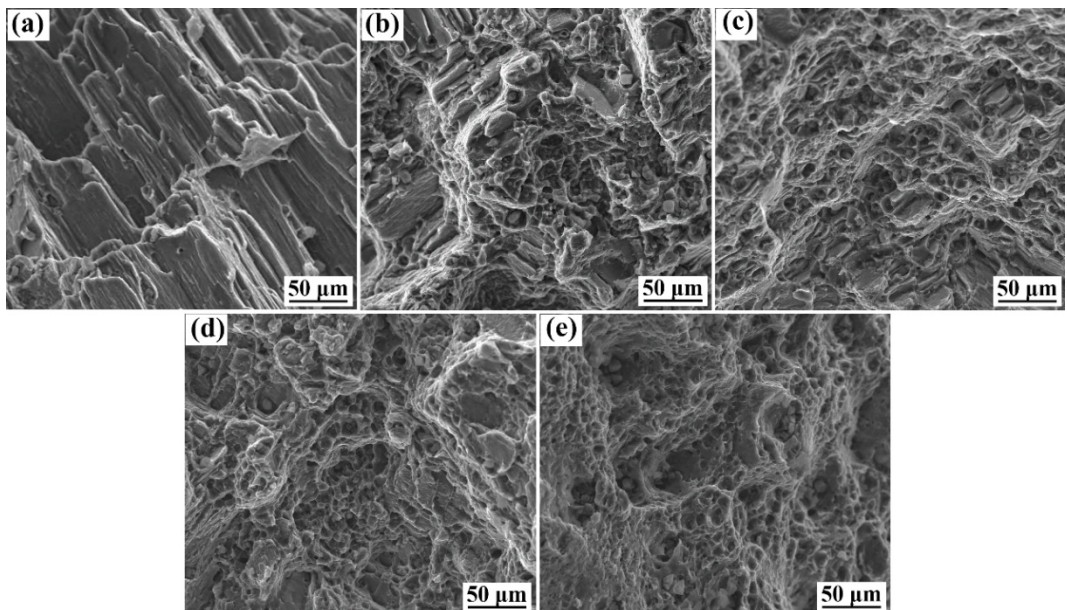

**Figure 6.** Fracture morphologies of as-homogenized Mg-*x*Li-3Al-2Zn-0.2Zr alloys: (**a**) *x* = 5; (**b**) *x* = 7; (**c**) *x* = 8; (**d**) *x* = 9; (**e**) *x* = 11.

Figure 7 depicts the side surfaces of the fracture for as-homogenized Mg-*x*Li-3Al-2Zn-0.2Zr alloys. In as-homogenized LAZ532-0.2Zr and LAZ732-0.2Zr alloys, the microvoids mainly came from the rupture of AlLi particles with larger sizes (marked by a red oval), and large microcracks could even be observed in the α-Mg matrix (marked by a yellow

oval), while no microvoids were found around the fibrous AlLi phase and tiny AlLi particles. In the as-homogenized LAZ832-0.2Zr alloy, the microvoids mainly came from the separation of AlLi particles from the matrix, and in addition the separation of the matrix phase interface (marked by a blue oval) due to the AlLi particles was extremely tiny. In as-homogenized LAZ932-0.2Zr and LAZ1132-0.2Zr alloys, microvoids caused by the rupture of AlLi particles and the separation of AlLi particles from the matrix were observed. Since the fracture mechanism of the α-Mg matrix is mainly cleavage fracture, large and straight microcracks could be observed in the α-Mg matrix. These microcracks grew through the grains and continued to expand, and finally showed the cleavage steps. The fracture mechanism of the β-Li matrix was a microvoid coalescence fracture with the basic feature of dimples, and the microvoids were formed by the rupture of the AlLi phase and the separation of the phase interface. According to statistics, AlLi particles with a diameter greater than 1.55 μm break easily during the tensile test, the smaller particles are prone to separate from the matrix, and extremely tiny particles cannot produce microvoids easily. The formation of microvoids at tiny particles is more difficult than at large particles, because the tensile stress required for the formation of microvoids is inversely proportional to the square root of particle size. Therefore, except for the fact that the increase of the β-Li matrix greatly improves the alloy elongation, tiny AlLi particles inhibit the formation of microvoids and are also conducive to the improvement of elongation.

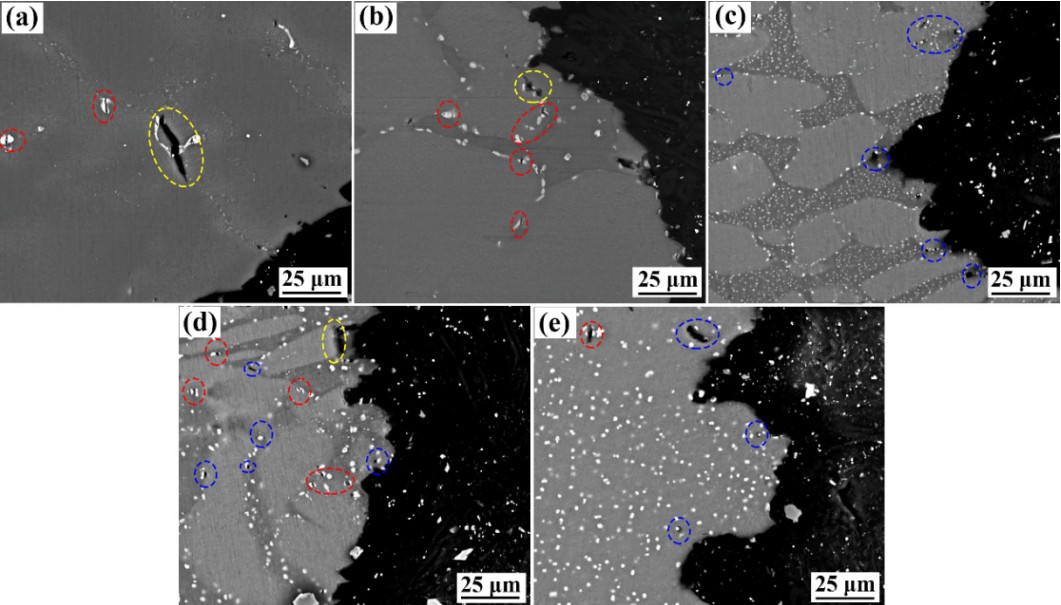

**Figure 7.** Side surfaces of the fracture for as-homogenized Mg-*x*Li-3Al-2Zn-0.2Zr alloys: (**a**) *x* = 5; (**b**) *x* = 7; (**c**) *x* = 8; (**d**) *x* = 9; (**e**) *x* = 11.

## 4. Conclusions

(1) When the Li content increases from 5 wt.% to 11 wt.%, the alloy matrix changes from the α-Mg single-phase to the α-Mg+β-Li dual-phase, and then to the β-Li single-phase. All alloys contain the AlLi phase, and its content gradually increases with the increase in Li content;

(2) Part of the Al element in the alloys forms the AlLi phase, and the remainder is mainly dissolved in the α-Mg matrix. Zn element in the alloys does not form compounds; part of Zn is enriched in the AlLi phase, and the rest is dissolved in the α-Mg matrix and β-Li matrix;

(3) As the Li content increases, the tensile strength and hardness of as-homogenized Mg-*x*Li-3Al-2Zn-0.2Zr alloys decrease while the elongation increases, and the corresponding fracture mode changes from cleavage fracture to microvoid coalescence fracture. This is mainly ascribed to the matrix of alloys changing from α-Mg with an hcp structure to β-Li with a bcc structure.

**Author Contributions:** Conceptualization, G.S. and Y.S.; methodology, Y.S.; validation, F.Z.; formal analysis, Y.S. and J.R.; investigation, Y.S., F.Z. and Y.S.; resources, J.R.; data curation, F.Z.; writing—original draft preparation, Y.S. and J.R.; writing—review and editing, J.R. and G.S.; supervision, G.S.; funding acquisition, Y.S. and J.R. All authors have read and agreed to the published version of the manuscript.

**Funding:** The authors are grateful for the financial support from the open project of the Key Laboratory of Green Fabrication and Surface Technology of Advanced Metal Materials (No. GFST2021KF04 and No. GFST2021KF09), the University Natural Science Research Project of Anhui Province (No. KJ2021A0394 and No. KJ2021A0395), and Anhui Provincial Natural Science Foundation (No. 2208085QE124).

**Data Availability Statement:** The raw data required to reproduce these findings cannot be shared at this time as the data also forms part of an ongoing study.

**Conflicts of Interest:** The authors declare that they have no known competing financial interest or personal relationships that could have appeared to influence the work reported in this paper.

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
