# Peer review of "Effect of Li Content on the Microstructure and Mechanical Properties of as-Homogenized Mg-Li-Al-Zn-Zr Alloys"

_alloys, doi:10.3390/alloys2020006_

Round 1
Reviewer 1 Report
This paper describes a study of a series of alloys based on the Mg-Li systems with additions of Zn and Al. This class of alloy has been subject to intensive investigation previously due to its ultra-low density and other attractive properties. The present study builds on this work and so is useful to the scientific community, but does not provide significant new insights into the fundamentals of such materials. The paper is generally well written, but there are some minor errors to be corrected:
1. Check English carefully (e.g. "alloys ... were" not "alloys ... was"
2. The description of a particle morphology as "filamentous" is non-standard and not an accurate description of what is seen in the images. These particles seem to be either rod shaped or plates (viewed edge on). The authors should clarify the true morphology (e.g. through tilting in the TEM) and correct throughout
Optional revision - the paper presents only 2 TEM images of one condition in one alloy. It would be of great value to improve the paper by showing example TEM images of particle morphologies in the other alloys.
Reviewer 2 Report
Results are stimulating. In order to improve the puplication, it is required to plot the stress-strain curves. In addition authors should present the properties divided by the density.
Reviewer 3 Report
The manuscript can not be accepted in its present form due to the following major points:
1. The main point is not clear concerning the type and application of alloys studied. The new in in this manuscript is missing relating to dozen of articles published in same point. The authors should make their new research point more clear and correlate it with has been published before.
2. The experimental section has to revised carefully. There are missing data that should be mentioned as thermal treatment parameters. More discussion about the objective and vision should be added. The experimental steps should be added in more details.
Round 2
Reviewer 3 Report
The authors did most of required revision and modification. The manuscript can be accepted in its present form.